# Forecasting Wastewater Temperature Based on Artificial Neural Network (ANN) Technique and Monte Carlo Sensitivity Analysis

**Farzin Golzar** [1,*] , **David Nilsson** [2] **and Viktoria Martin** [1]

[1] Division of Energy Systems, Department of Energy Technology, KTH-Royal Institute of Technology, 11428 Stockholm, Sweden; viktoria.martin@energy.kth.se
[2] Water Centre, KTH Royal Institute of Technology, 11428 Stockholm, Sweden; david.nilsson@abe.kth.se
* Correspondence: fargo@kth.se; Tel.: +46-8790-7441

**Abstract:** Wastewater contains considerable amounts of thermal energy. Heat recovery from wastewater in buildings could supply cities with an additional source of renewable energy. However, variations in wastewater temperature influence the performance of the wastewater treatment plant. Thus, the treatment is negatively affected by heat recovery upstream of the plant. Therefore, it is necessary to develop more accurate models of the wastewater temperature variations. In this work, a computational model based on artificial neural network (ANN) is proposed to calculate wastewater treatment plant influent temperature concerning ambient temperature, building effluent temperature and flowrate, stormwater flowrate, infiltration flowrate, the hour of day, and the day of year. Historical data related to the Stockholm wastewater system are implemented in MATLAB software to drive the model. The comparison of calculated and observed data indicated a negligible error. The main advantage of this ANN model is that it only uses historical data commonly recorded, without any requirements of field measurements for intricate heat transfer models. Moreover, Monte Carlo sensitivity analysis determined the most influential parameters during different seasons of the year. Finally, it was shown that installing heat exchangers in 40% of buildings would reduce 203 GWh year$^{-1}$ heat loss in the sewage network. However, heat demand in WWTP would be increased by 0.71 GWh year$^{-1}$, and the district heating company would recover 176 GWh year$^{-1}$ less heat from treated water.

**Keywords:** heat recovery; artificial neural network technique; wastewater temperature; sewer; Monte Carlo simulation; Stockholm

## 1. Introduction

To tackle climate change impacts, several international and national targets have been determined. In December 2015, almost all the countries of the world agreed on the Paris Agreement. The main goal of the agreement is to strengthen the global response to the threat of climate change by keeping the global temperature increase at lower than 2 °C and to limit the rise even further to 1.5 °C, compared to pre-industrial levels [1]. The agreement states the need for all countries to set climate goals. To fulfil these ambitious goals, applicable evolution and arrangement of financial resources, innovative technology framework, and improved capacity-expansion are to be put in place [1]. In 2015, the UN also adopted Agenda 2030, consisting of 17 global goals for sustainable development to achieve a better and more sustainable future for all [2]. The European Union has undertaken quantitative targets of decreasing greenhouse gas emissions by 40% compared to the level in 1990 until 2030, as well as a 27% increase in the share of renewable energy and 7% enhanced energy efficiency [3]. The Energy

Performance of Buildings Directive (EPBD) addresses the contribution of buildings; the sector accounts for 40% of energy use and 36% of $CO_2$ [4]. The EU aims at all new buildings being nearly-zero-energy buildings, while renovation strategies shall improve the performance of old buildings [5]. As part of the EU, Sweden's ambition is to be a leading country in the achievement of the objectives of the Paris Agreement and EU targets. Moreover, Sweden's long term target is net-zero carbon emissions by 2045, and thereafter negative emissions [6]. In this paper, we explore how wastewater heat recovery can contribute to climate goals. Specifically, our objective is to present and evaluate a method for forecasting water temperature at the wastewater treatment plant level, which is an essential parameter for assessing and designing heat recovery systems on a larger scale.

Heating and cooling of buildings have been considered as one of the areas where the use of renewable energy is substantially improved [7,8]. A technology that is promising regarding this is recovering the thermal energy contained in wastewater and using it for heating [9–15]. Studies investigated that if wastewater were cooled down by 1 °C, 720 GWh year$^{-1}$ would be theoretically gained [16]. More than 500 existing wastewater heat recovery systems have been reported all around the world [17], which are mostly located in Switzerland, Germany, and Scandinavia [18]. Some countries like Austria include the thermal energy of wastewater as a renewable energy resource in their national energy policies [17]. Therefore, it is expected that wastewater heat recovery technologies such as heat exchangers and heat pumps will be further expanded in the upcoming years.

Generally, there are two approaches for wastewater heat recovery [17]. The first one is the implementation of heat recovery technologies after wastewater treatment plant (WWTP) to recover heat from treated water [17,19]. Small fluctuations in flow and temperature of WWTP effluent is beneficial in heat pumps. Furthermore, treated water causes less fouling and clogging problems in wastewater heat recovery (WWHR) technologies. More importantly, heat recovery from treated water does not affect the biological performance of WWTP. That is why several WWTPs in Sweden were able to recover 2 to 3 GWh year$^{-1}$ of heat from treated water [20]. There are several studies about heat recovery from WWTP effluent in Switzerland [21], Russia [22], and Scandinavia [21,23], indicating the maturity of WWHR technologies for treated water. Hepbasli et al. [18] reviewed the performance of heat pumps installed at WWTPs from energetic, exergetic, environmental, and economic points of view. However, treated water has lost part of its thermal energy in the sewer network. This leads to a lower temperature, which influences the performance of any heat recovery system. Meggers and Leibundgut [24] reported that the coefficient of performance (COP) of heat pumps could be reduced from 7 at buildings to 3 at WWTPs.

The second approach is the installation of heat recovery technologies such as heat exchangers or heat pumps at the property level [8,25]. Heat recovery at the property level and before WWTP is encouraging as wastewater contains more thermal energy than treated water. Another advantage of property level heat recovery is that recovered heat is used in very close vicinity to the WWHR system by short transport distances, which is essential in terms of economical implementation and operation. Frijns et al. [15] showed that 21.6 MWh year$^{-1}$ theoretical heat could be recovered from wastewater in Dutch households. Alnahhal and Spremberg [25] investigated that greywater heat recovery could fulfil 30% of domestic hot water demand in buildings. Several studies evaluated the energy and economic performance of heat exchangers [24,26–32] or combined systems of heat exchangers and heat pumps [33–38] for greywater heat recovery. Lin et al. [39] improved the performance of heat recovery technologies using heat pipes. Many recent studies focused on particular applications such as dishwashers [40–46], and washing machines [41,42,46,47]. Wärf et al. [48] concluded that the potential heat recovery and maximum temperature drop associated with heat recovery on a case study in Linköping, Sweden is 0.65 kWh person$^{-1}$ day$^{-1}$ and 4.2 °C.

Nevertheless, the major drawback of on-property level heat recovery is the negative effects on the temperature-dependent processes in WWTP due to the cooling down of wastewater, which may lead, for example, to sub-optimal biological processes within the treatment plant [49]. Therefore, plans for upscaling property-level wastewater heat recoveries must not only investigate energy economy aspects,

but also consider wastewater temperature related issues. Particularly in most countries, some legal constraints are determined on the temperature reduction for influents of WWTPs [16,49]. Therefore, a comprehensive understanding of wastewater temperature development, as a function of the expansion of on-property heat recovery, is essential.

A number of studies have developed various models for temperature changes in the wastewater system with different degrees of complexity. Sonakiya et al. [50] and Abdel-Aal et al. [51] proposed linear models of temperature loss as a function of sewer length. Since the model depends on the sewer length, the accuracy of the model varies for different sites at the sewer pipe. For example, Abdel-Aal et al. [51] reported their model error, which is the difference between modelled and measured temperature drops, in the range of −1 K to 0.76 K in different points of sewage network. Therefore, these linear models are associated with substantial uncertainty [19].

On the other hand, Dürrenmatt and Wanner [16,52] described a detailed model entitled TEMPEST by considering all relevant energy and mass flows. TEMPEST depends on a wide variety of parameters related to the sewer pipe, soil, wastewater, and air, which some of the sensitive parameters should be calibrated by the use of field measurement data. Abdel-Aal et al. [51] argued that the requirement of many details and parameters about the sewer system makes TEMPEST infeasible to implement. Moreover, for investigating the impacts of local heat recoveries on the WWTP influent temperature, simulating detailed processes in the sewage network, which adds the complexity of the model, is unnecessary. Kretschmer et al. [17] presented a more straightforward model by focusing on the maximum potential of heat recovery in the sewer network. Nonetheless, this model does not investigate the potential influence of wastewater heat recovery on heat losses in the sewer system [19]. Literature review shows that there is a conflict between the accuracy and simplicity of wastewater temperature models.

Artificial intelligence (AI) techniques are practical methods in overcoming such conflicts. AI is proven to be implemented as an alternative to process-driven physical models due to no need for detailed knowledge of internal system parameters [53,54]. Wei et al. [55] reviewed and compared conventional models and AI-based models implemented for energy consumption over the past decades. They concluded that AI-based models are reliable and full-scale in forecasting horizons. Bylinski et al. [56] concluded that using an artificial neural network (ANN) grants a great reflection of complex dependencies of the wastewater management problems, without considering detailed mechanisms of specified processes.

In this paper, an ANN-based model is developed to forecast the dynamic behaviour of wastewater temperature at the entrance of WWTP as a function of ambient temperature, building effluent temperature and flowrate, stormwater flowrate, infiltration flowrate, the hour of day, and the day of year. In addition to the fact that ANN does not need detailed knowledge of internal system processes, we use historical data commonly recorded at WWTP. The model is implemented in MATLAB software, version 9.7 (The MathWorks, Natick, MA, USA), and the formulation between inputs and output is presented. The model is trained by historical data from 2009 to 2018 and validated by 2019 data obtained from Henriksdal WWTP in Stockholm. Furthermore, the impact of uncertain input parameters on the ANN model output is scrutinised using the Monte Carlo simulation (MCS) technique. Finally, the proposed model is implemented to investigate the impacts of upscaling local heat recoveries on the performance of wastewater systems in Stockholm. MATLAB software has been employed for developing the artificial neural network (ANN), statistical analysis, and Monte Carlo simulation.

## 2. Materials and Methods

A forecasting model should have the flexibility and capability to deal with data accurately [57]. In this work, the ANN technique is selected due to no need for detailed knowledge of internal sewage network parameters. This technique needs historical data commonly recorded, without any requirements of field measurements for intricate heat transfer models. Moreover, the ANN is used due to its ability to train, validate, and test its parameters [58]. ANN models are considered as a non-linear

modelling technique, which facilitates the formulation of links among input and output parameters by adequate weights and activation functions. In addition, a Monte Carlo sensitivity analysis is selected to allocate the uncertainty in the output of ANN model to the uncertainty associated with each input parameter over their entire range of interest. Monte Carlo sensitivity analysis changes all the input parameters simultaneously, which causes it to determine the most influential input parameters during the year and also during different seasons of the year. Different steps in constructing the proposed forecasting model are explained as follows.

*2.1. Input Data*

The first step in the ANN process is the determination of ANN model inputs. To realise impactful parameters, *WWTP influent temperature*, WWTP influent flow rate, and ambient temperature have been investigated and compared in Figure 1, based on real data as mentioned. Ambient temperature has a seasonal impact on *WWTP influent temperature*. For instance, in cold seasons where ambient temperature is in the range of −10 to 10 °C, *WWTP influent temperature* is in the range of 8 to 15 °C. On the other hand, in hot seasons where ambient temperature increases to the range of 10 to 25 °C, *WWTP influent temperature* is in the range of 15 to 20 °C. Wastewater leaving buildings has an average temperature of around 25 °C all around the year [15]. Therefore, the seasonal variation of *WWTP influent temperature* is a function of ambient temperature. However, the temperature of wastewater leaving buildings fluctuates during the day and affects diurnal variations of *WWTP influent temperature*. Furthermore, Figure 1 shows that whenever there is a remarkable rise in WWTP influent flowrate, *WWTP influent temperature* decreases substantially. It indicates that hourly fluctuations in *WWTP influent temperature* are a function of flowrate. The wastewater enters WWTP contents in three main components: (i) Wastewater from buildings, (ii) water infiltration into the sewage system, and (iii) stormwater. The amount of wastewater from buildings is in the range of 0.0026 to 0.0084 m$^3$ h$^{-1}$ pe$^{-1}$ [59]. There are 752,700 people connected to Henriksdal WWTP in Stockholm [60]. Thus, the flow rate of wastewater dispatched from buildings is in the range of 1957 to 6322 m$^3$ h$^{-1}$. The amount of infiltration to the sewage system is simulated as a sine wave with the lowest values during the dry season and the highest during the rainy season [61]. The amount of stormwater enters Henriksdal WWTP is quantified by subtracting the amount of wastewater leaving buildings and infiltration from the total amount of WWTP influent flowrate. Finally, ambient temperature, building effluent temperature, building effluent flowrate, stormwater flowrate, infiltration flowrate into the sewage network, time of day, and day of year are chosen as input variables.

The data series for ambient temperature is received from the Swedish Meteorological and Hydrological Institute (SMHI) [62]. The data series for *WWTP influent temperature* and flowrate are received from Stockholm Vatten och Avfall AB [63]. Building effluent temperature and flowrate have not been continuously measured. However, data from random pumping stations in Stockholm shows that building effluent temperature and flowrate have constant trends with periodic daily fluctuations, as discussed by Cipolla and Maglionico [64]. Therefore, the data series for building effluent temperature is calculated based on the average value of 25 °C [15] and the periodic daily fluctuations adapted from pumping stations data. Data series for building effluent flowrate simulated based on the daily lowest and highest values of 5018 m$^3$ h$^{-1}$ and 6322 m$^3$ h$^{-1}$ and daily fluctuations of wastewater passing through pumping stations. The minimum amount of flowrate during summer is decreased to 1957 m$^3$ h$^{-1}$ due to travelling. Table 1 summarises input and output variables and their minimum and maximum values from 2009 to 2019, and the complete dataset can be found in the supplementary data file available online.

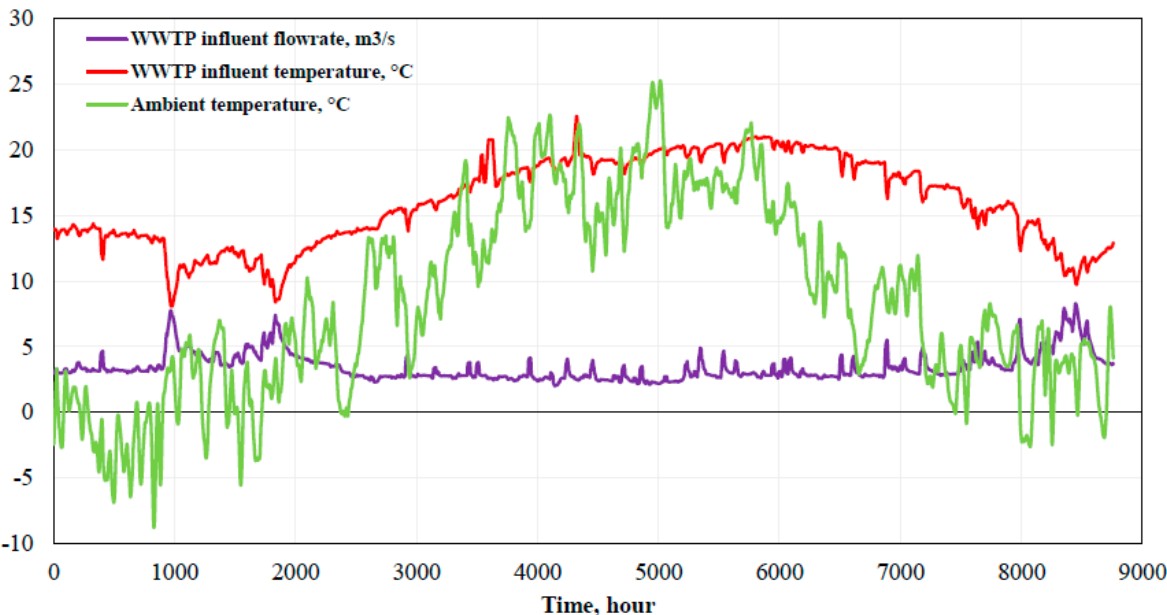

**Figure 1.** The variation of wastewater treatment plant (WWTP) influent temperature as a function of WWTP influent flowrate and ambient temperature. Data from Stockholm Vatten och Avfall AB [64] and the Swedish Meteorological and Hydrological Institute (SMHI) [63].

**Table 1.** The minimum and maximum of input and output variables for the city of Stockholm, Henriksdal WWTP from 2009 to 2019.

| Set | Parameter | Unit | Minimum | Maximum |
|---|---|---|---|---|
| | Ambient temperature | °C | −20.70 | 31.80 |
| | Buildings effluent temperature | °C | 8.51 | 34.70 |
| | Buildings effluent flowrate | $m^3 s^{-1}$ | 0.55 | 1.74 |
| Inputs | Stormwater flowrate | $m^3 s^{-1}$ | 0.00 | 11.08 |
| | Infiltration flowrate | $m^3 s^{-1}$ | 0.00 | 0.58 |
| | Hour of day | h | 1.00 | 24 |
| | Day of year | d | 1.00 | 365 |
| Outputs | *WWTP influent temperature* | °C | 2.08 | 23.70 |

### 2.2. ANN Procedure

The structure depicted in Figure 2, defined as an ANN model, includes three main layers: (i) Input layer, (ii) hidden layers, and (iii) output layers. Although only one hidden layer and one output are presented in Figure 1 as a general demonstration, there could be more than one hidden layer and several outputs. Input ($I_i$) is multiplied by weight ($W^I_{i,j}$) and are summed up with biases in the hidden layer ($b^H_j$) and are collected in the hidden layer as the neurons ($N_j$). Afterwards, the obtained values are transferred to the output layer by implementing a transfer function ($f^H$) and multiplied into hidden layer's weights ($W^H_{1,j}$). Finally, the summation of output layer bias provides the final value of the ANN model output.

To prepare the ANN model, the dataset has to be divided into three sets as training (70% of available data), validation (15% of available data), and testing (15% of available data) [58]. By generating initial weights and biases, which are random numbers close to zero, the training stage started. By implementing an optimisation algorithm for regulating weights to minimise the error between the simulated output and the observed output, the network is trained. In this research, Levenberg-Marquardt backpropagation (trainlm) was used as the training algorithm. Then, the regulated weights and biases of hidden and output layers are implemented to validate the results, based on the defined observed

data. If the error were not satisfactory, the weights and biases would change until the error between the simulated output and the observed output in the training and validation sets became agreeable [65]. After training and validation stages, the adjusted weights and biases are used to calculate outputs considering the dataset dedicated to the testing stage. It is crucial not to use the data points of training and validation stages in the testing stage. That is why the proposed ANN model would be able to be used as an extrapolation-forecasting tool. In other words, it would be able to forecast output parameters based on the data, which is not included in the training and constructing stages. In this work, MATLAB 2019 is employed for simulating the system and also for data processing.

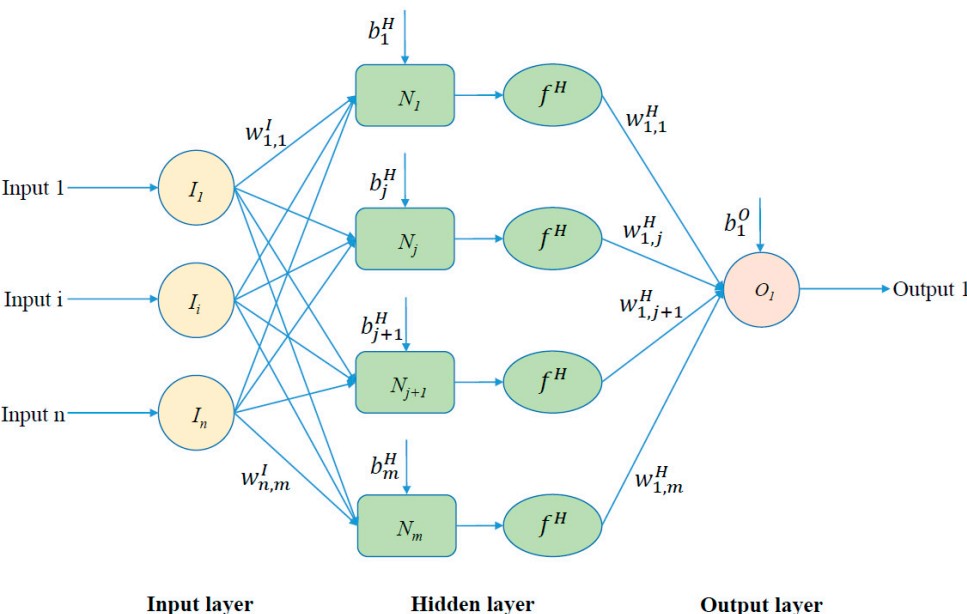

**Figure 2.** Schematic representation of the artificial neural network (ANN) structure.

Since the order of input variables are dissimilar (see Table 1), they should be normalized between $-1$ and 1 or 0 and 1 before the training stage [57,66]. Therefore, actual data ($X_i$) were normalized ($Y_i$) in the range of $-1$ and 1 using the following Equation (1):

$$Y_i = 2 \times \left( \frac{X_i - X_{min}}{X_{max} - X_{min}} \right) - 1 \tag{1}$$

in which $X_{max}$ and $X_{min}$ are maximum and minimum values of variable *i*.

### 2.3. ANN Architecture

For the aim of finding the optimum ANN structure, several parameters should be adjusted. The most influential parameters in ANN are the number of hidden layers, the number of neurons in each hidden layer, and the transfer function [66]. A trial-and-error method is implemented to determine the optimum values of the aforementioned parameters with the objective of minimum error. In this work, the coefficient of determination ($R^2$), relative root mean square error (*RRMSE*), and percent bias (*PBIAS*) are chosen as error indicators. $R^2$ calculated by Equation (2), based on Cheng et al. [67], evaluates the degree of the linear relationship between simulated and observed data. *RRMSE*, calculated by Equation (3) based on Sirsat et al. [68], grants a relative model evaluation, and its value is in the range of 0 and 1. *RRMSE* = 0 being the optimal value and the lower values showing the less error [68]. *PBIAS* determines the tendency of the simulated data to overestimate or underestimate the observed data [69]. Positive values indicate model underestimation bias, while

negative values present model overestimation bias [70]. *PBIAS* is calculated with Equation (4) based on Gputa et al. [70].

$$R^2 = 1 - \left( \frac{\sum_{i=1}^{n} \left( X_i^{obs} - X_i^{sim} \right)^2}{\sum_{i=1}^{n} \left( X_i^{obs} - X_i^{mean} \right)^2} \right) \tag{2}$$

$$RRMSE = \frac{1}{X^{mean}} \times \sqrt{\frac{1}{n} \sum_{i=1}^{n} \left( X_i^{obs} - X_i^{sim} \right)^2} \tag{3}$$

$$PBIAS = \left( \frac{\sum_{i=1}^{n} \left( X_i^{obs} - X_i^{sim} \right)}{\sum_{i=1}^{n} \left( X_i^{obs} \right)} \right) \tag{4}$$

where $X_i^{obs}$ is the observed data, $X_i^{sim}$ is the simulated value, $X_i^{mean}$ is the mean of observed data, and $n$ is the total number of observations.

### 2.4. Monte Carlo Simulation (MCS)

In Monte Carlo simulations (MCS), random samples of uncertain input parameters were generated regarding pre-defined probability density functions (PDFs). In this paper, probability distributions were defined based on historical data and tested using Kolmogorov-Smirnov test (KS test) [71,72]. In other words, we assumed a PDF for each input parameter and evaluated the goodness-of-fit of the assumed PDF by the KS test. This test includes two parameters as the KS test statistic and KS test *p*-value. KS test statistic represents the maximum distance between the cumulative density functions (CDFs) of assumed distribution and samples of the input parameter. The more this value is close to zero, the more likely that assumed distribution fits the observed data. If the *p*-value is more than the significance level (0.05 in this work), it is accepted that the assumed distribution fits the observed data [73].

The randomly generated samples from input PDFs were implemented as the inputs of the developed ANN model, and the output for each input vector was calculated. The outputs were used to determine the correlation between uncertain input parameters and output.

## 3. Results and Discussion

### 3.1. Developed Model Description

In order to determine the optimum structure of the ANN model, we used historical data from 2009 to 2018 to train several structures with a different number of hidden layers (one or two), a different number of neurons in each layer (0 to 10), as well as various transfer functions. The zero (0) value for the number of neurons in the second hidden layer means that the network only has one hidden layer. The errors associated with 20 structures are summarised and compared in Table 2. The results show that the best structure would be an input layer with 7 input variables, 1 hidden layer with 10 neurons, and an output layer with 1 output variable. Therefore, the optimum structure is presented as 7-10-1. Also, six linear and non-linear transfer functions commonly used in ANNs techniques are randomly tested. Table 2 indicates that 7-10-1 structure with Tan-sigmoid as transfer function provides the highest $R^2$ and lowest *RRMSE* and *PBIAS* for the outputs. Figure 3 demonstrates the optimum structure of ANN network, which has 7 input parameters, 1 hidden layer with 10 neurons, 1 output layer with 1 output variable, and Tan-sigmoid as transfer function. The optimum values of weights and biases related to input and hidden layer are reported in Table 3. Thus, considering the optimum

weights and biases in Table 3 and Equation (5), *WWTP influent temperature* would be calculated for each input dataset.

$$WWTP\ Influent\ Temperature = W_{1,j}^H \times \left( \left( \frac{2}{1 + exp\left(-2 \times \left(\left(W_{i,j}^I\right)' \times X\right) + b_j^H\right)} \right) - 1 \right) + b_1^O \qquad (5)$$

$i$ = input counter, $j$ = neuron counter.

**Table 2.** Comparison of different neural networks with two hidden layers and varied number of neurons as well as different transfer functions.

| No. | Structure | WWTP Influent Temperature | | | |
| | | *Transfer Function* | $R^2$ | *RRMSE* | *PBIAS* |
|-----|-----------|---------------------------|-------|---------|---------|
| 1 | 7-4-0-1 | Symmetric hard-limit | 0.845 | 0.071 | 0.056 |
| 2 | 7-4-3-1 | Triangular Basis | 0.932 | 0.059 | 0.107 |
| 3 | 7-4-6-1 | Tan-sigmoid | 0.897 | 0.065 | 0.043 |
| 4 | 7-5-1-1 | Hard-limit | 0.963 | 0.048 | 0.090 |
| 5 | 7-5-5-1 | Satlin | 0.920 | 0.050 | 0.194 |
| 6 | 7-5-7-1 | Symmetric hard-limit | 0.863 | 0.062 | 0.036 |
| 7 | 7-6-0-1 | Satlin | 0.864 | 0.059 | 0.039 |
| 8 | 7-6-3-1 | Triangular Basis | 0.967 | 0.053 | 0.164 |
| 9 | 7-6-5-1 | Linear | 0.942 | 0.055 | 0.072 |
| 10 | 7-7-0-1 | Hard-limit | 0.852 | 0.072 | 0.116 |
| 11 | 7-7-4-1 | Satlin | 0.914 | 0.056 | 0.087 |
| 12 | 7-7-8-1 | Linear | 0.923 | 0.052 | 0.094 |
| 13 | 7-8-2-1 | Satlin | 0.901 | 0.058 | 0.073 |
| 14 | 7-8-4-1 | Triangular Basis | 0.864 | 0.076 | 0.036 |
| 15 | 7-8-8-1 | Symmetric hard-limit | 0.887 | 0.063 | 0.059 |
| 16 | 7-9-0-1 | Hard-limit | 0.854 | 0.059 | 0.061 |
| 17 | 7-9-4-1 | Satlin | 0.935 | 0.052 | 0.081 |
| 18 | 7-9-7-1 | Symmetric hard-limit | 0.918 | 0.057 | 0.077 |
| 19 | 7-9-10-1 | Linear | 0.965 | 0.049 | 0.067 |
| 20 | **7-10-0-1** | **Tan-sigmoid** | **0.983** | **0.044** | **0.035** |

**Table 3.** The optimum weights ($w_{ij}$) and biases (bj) for the best architecture of ANN models for predicting *WWTP influent temperature*.

| Layer | Weight and Bias | No. | No. of Neurons | | | | | | | | | |
| | | | 1 | 2 | 3 | 4 | 5 | 6 | 7 | 8 | 9 | 10 |
|-------|-----------------|-----|------|-------|-------|-------|-------|-------|-------|-------|-------|-------|
| Input | Weights, $w_{i,j}^I$ | 1 | 0.18 | −2.61 | 0.65 | 0.68 | 5.83 | 4.54 | 0.12 | 0.42 | −0.08 | −7.83 |
| | | 2 | 0.20 | 0.78 | 0.00 | −3.01 | −0.05 | 0.09 | −0.04 | −3.74 | −4.43 | 0.07 |
| | | 3 | 0.45 | −0.76 | −0.04 | 0.09 | 0.00 | 0.18 | 0.42 | 0.05 | −4.54 | 0.04 |
| | | 4 | −0.32 | −3.04 | −0.06 | −0.26 | 0.00 | 0.10 | 0.03 | −0.45 | 1.10 | 0.00 |
| | | 5 | 0.36 | 0.27 | 0.00 | −0.09 | 0.01 | 0.10 | 0.11 | −0.06 | −0.28 | −0.01 |
| | | 6 | 3.08 | −1.14 | 0.13 | 0.33 | 0.25 | 0.14 | 0.25 | 0.58 | −0.50 | −0.71 |
| | | 7 | 3.03 | −1.11 | −0.17 | −0.68 | 0.22 | 1.03 | 1.94 | −0.96 | 0.10 | 0.14 |
| Hidden | Bias, $b_j^H$ | 1 | 3.97 | 2.07 | −0.87 | −0.9 | 1.15 | −0.08 | 1.33 | −1.17 | −1.34 | −1.34 |
| | Weights, $w_{1,j}^H$ | 1 | 0.13 | −0.09 | −3.11 | 1.08 | 3.02 | 0.18 | −0.53 | −1.03 | −0.02 | 2.11 |
| Output | Bias, $b_1^O$ | 1 | −1.86 | | | | | | | | | |

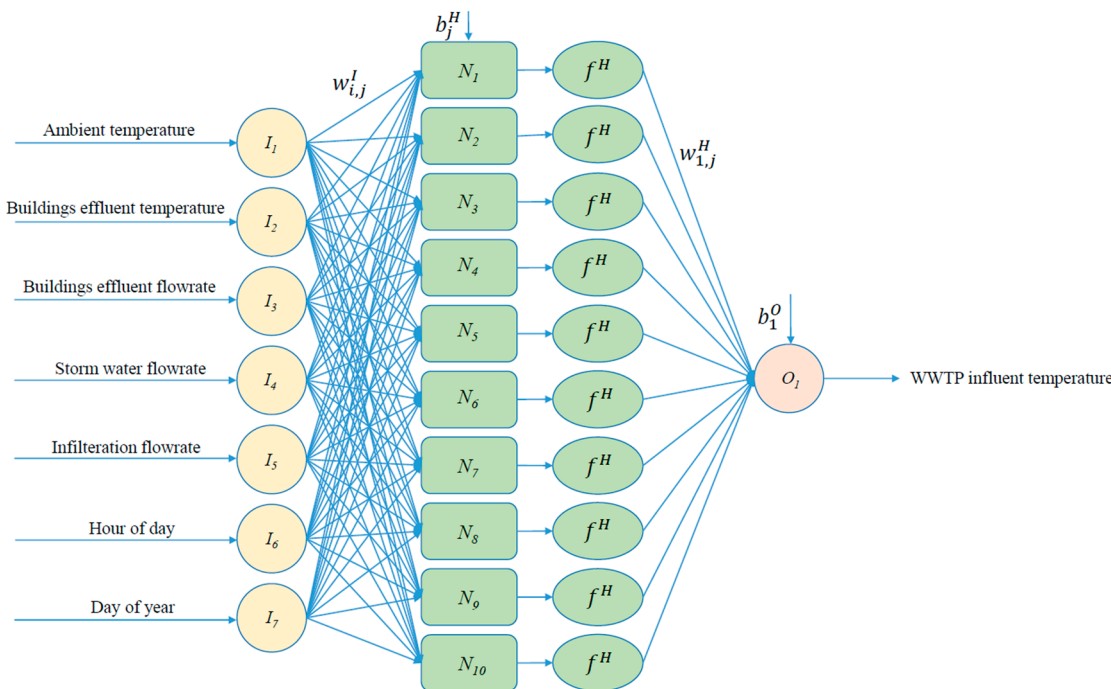

**Figure 3.** Optimum structure of ANN model for forecasting the *WWTP influent temperature*.

### 3.2. Testing and Validation of the Developed Network

Figure 4 shows parity plots for *WWTP influent temperature* calculated from the network against the observed data related to the years 2009 to 2018. The best fit of outputs versus targets was recognisized by a solid diagonal line while the ideal fit is represented by the dashed diagonal line. The deviation of the best fit (solid line) from the ideal fit (dashed line) shows the difference between correlated values and observed ones. Figure 4a represents the parity plots for the training stage. The correlation coefficient ($R^2$) for the training dataset is equal to 0.979, which shows that there is a good agreement between calculated and observed data. To validate and test the developed network, other datasets, which were not considered in the training stage, were taken into account. Figure 4b,c shows the correlation between calculated and observed data in validation and testing stages. The correlation coefficient ($R^2$) for validation and testing stages are 0.980 and 0.979, respectively. Figure 4d depicts that the aggregation of all points is located around the bisection, and this reveals the accuracy of the results and the ability of the proposed ANN models for forecasting *WWTP influent temperature*.

### 3.3. Extrapolation Capacity of the Developed Network

For evaluating the extrapolation ability of the developed network, the dataset related to the year 2019 was prepared as input to the network. These collected data were not used in training, testing, and validation stages. Figure 5 compares the observed data and the network output (*WWTP influent temperature*) values for the year 2019. The correlation coefficient ($R^2$) value of 0.945 shows a highly linear relationship between observed and simulated data. The relative root mean square error (*RRMSE*) value of 0.055 reveals an acceptable deviation of simulated data from observed data. Moreover, a percent bias (*PBIAS*) value of 0.080 suggests an underestimation of the ANN model. Part of these errors is due to the deviation of simulated values from observed data during hours 3521 to 3535, 3571 to 3629, and 4298 to 4326. For instance, the reported value of the observed data for all hours in the first period is 20.94 °C, while the average value of simulated data for these hours is 17.52 °C. This comparison shows that simulated data are more logical. Experts from Henriksdal WWTP believe that due to sensor malfunction, the reported data in these periods are not correct. Generally, statistical indicators confirm an appropriate extrapolation ability of the developed network.

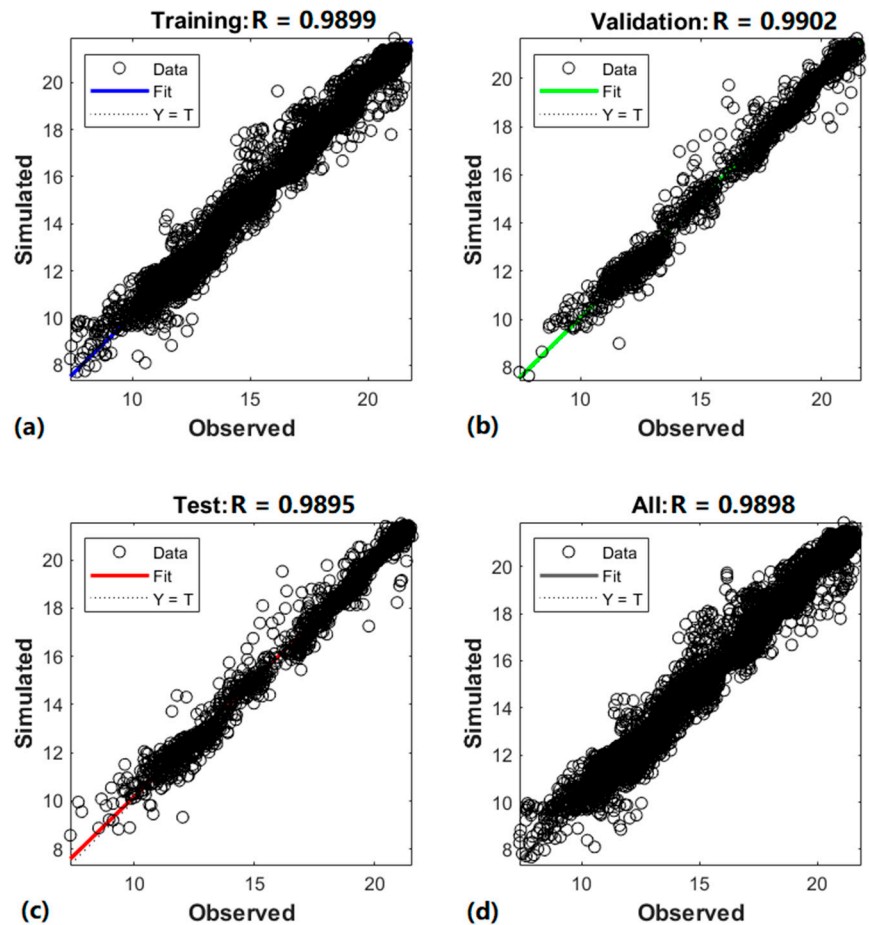

**Figure 4.** Correlation between values of calculated values and observed data for (**a**) training dataset, (**b**) validation dataset, (**c**) testing dataset, and (**d**) all datasets.

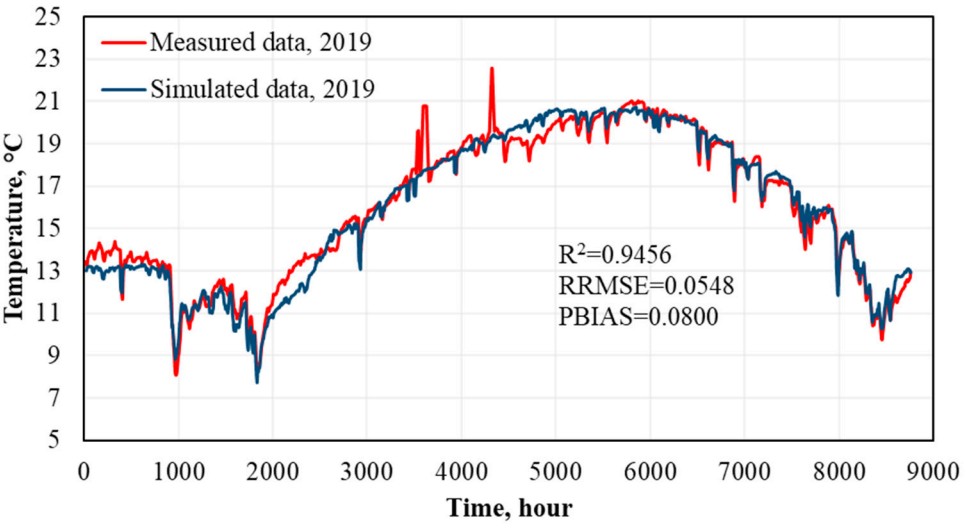

**Figure 5.** Measured (red) and simulated (blue) data for *WWTP influent temperature* for the year 2019.

*3.4. Sensitivity Analysis*

To conduct the Monte Carlo simulation (MCS) sensitivity analysis, the probability density functions (PDFs) of the uncertain input variables should be determined. Ambient temperature, building effluent temperature, building effluent flowrate, infiltration flowrate, and stormwater flowrate are uncertain

input parameters during the year. The assumed PDFs for inputs, as well as the KS test statistics and p-values, are presented in Table 4.

**Table 4.** Simulated probability density functions (PDFs) with the input ranges and obtained *WWTP influent temperature* (output) ranges.

| Uncertain Input Parameter | Unit | PDF | Scenario Range | Kolmogorov–Smirnov Test | | Δ Input | Δ Output (°C) |
| --- | --- | --- | --- | --- | --- | --- | --- |
| | | | | Statistic | *p*-value | | |
| Ambient temperature | °C | Gaussian mixture | $\mu_1 = 3.5; \sigma = 3.0$ $\mu_2 = 15.5; \sigma = 5.0$ | 0.02 | 0.45 | 42.60 | 11.74 |
| Buildings effluent temperature | °C | Normal | $\mu_1 = 22.33; \sigma = 2.95$ | 0.01 | 0.86 | 22.1 | 13.00 |
| Buildings effluent flowrate | m$^3$ s$^{-1}$ | Uniform | a = 1.20; b = 1.80 | 0.08 | 0.06 | 0.52 | 11.42 |
| Infiltration flowrate | m$^3$ s$^{-1}$ | Exponential | $\lambda = 0.09$ | 0.03 | 0.20 | 0.41 | 4.31 |
| Storm water flowrate | m$^3$ s$^{-1}$ | Exponential | $\lambda = 0.09$ | 0.01 | 0.70 | 7.93 | 14.59 |

For MCS, 5000 input vectors were sampled randomly regarding the selected PDFs (Table 4). The generated dataset was arranged with 5 blocks of 1000 vectors per input. In each block, only one input had random values in the defined range, while other inputs had mean values. The variation of inputs and accordingly obtained *WWTP influent temperatures* (output) are presented in Table 4.

The output ranges represent that stormwater flowrate and building effluent temperature are the most influential input parameters (impacts on output = 13 to 14.59 °C). The second most impactful parameters are ambient temperature and building effluent flowrate, of which their variations caused the output to change from 11.42 to 11.74 °C. The lowest influence is related to infiltration flowrate.

The standard deviation of *WWTP influent temperature* (output) that resulted from the variability of different inputs during different seasons of the year is presented in Figure 6. Figure 6a depicts the impacts of input parameters on *WWTP influent temperature* during winter. It shows that the stormwater flow rate is the most influential parameter in winter, while other parameters do not have considerable impacts on the *WWTP influent temperature*. However, Figure 6b illustrates that stormwater is not only not the dominant influential parameter in spring, but also that other parameters like building effluent temperature and ambient temperature are slightly more impactful. The reason is that stormwater in winter includes melted snow with a temperature around 0 °C. Thus, cold stormwater in winter has more impacts than warmer stormwater in spring. The variability of input parameters in spring is more than the rest of seasons, which is why the range of standard deviation of *WWTP influent temperature* in spring is more than other seasons. Figure 6c,d shows that building effluent temperature is the most influential input parameter during summer and autumn. However, the impacts of other input parameters like ambient temperature and building effluent flowrate are not ignorable. Generally, it is concluded that stormwater flowrate has remarkable effects on the temperature dynamics during winter, and other input parameters do not have considerable impacts on *WWTP influent temperature*. However, for the rest of the years, other input parameters like building effluent temperature, ambient temperature, and building effluent flowrate play a more critical role on the *WWTP influent temperature*.

For evaluating the uncertainty of *WWTP influent temperature*, another block of 1000 vectors as input data was sampled randomly regarding the PDFs determined for each input and introduced in Table 4. For each vector of input data, *WWTP influent temperature* was calculated, and a Gaussian mixture PDF with a *p*-value of 0.88 was fitted on the outputs. Figure 7 compares the PDF of observed data and simulated data. It demonstrates that the distribution of simulated data is very similar to the corresponding distribution of observed values.

*3.5. Model Application*

In the current situation, where there are not many local heat recoveries in the buildings in Stockholm, wastewater is discharged from buildings at 25 °C and received by WWTP at 16 °C on average. The developed model in this work is implemented to investigate the impact of local heat recoveries at the property level on inflow temperature to the treatment plant. It is assumed that

40% of the buildings connected to Henriksdal WWTP would install heat exchangers (HE) to recover heat directly in the buildings. Heat exchangers are passive heat recovery technologies, suitable for preheating cold water. Wastewater temperature drop in the heat exchangers depends on various parameters such as the type of heat exchanger, the flowrates and temperatures of wastewater as well as cold water, which vary during the day. Considering previous studies [48,64,74], 5 °C on average is assumed as the wastewater temperature drop in heat exchangers. A decrease of 5 °C in the temperature of discharged wastewater from 40% of buildings would lead to 2 °C decrease in the total building effluent temperature.

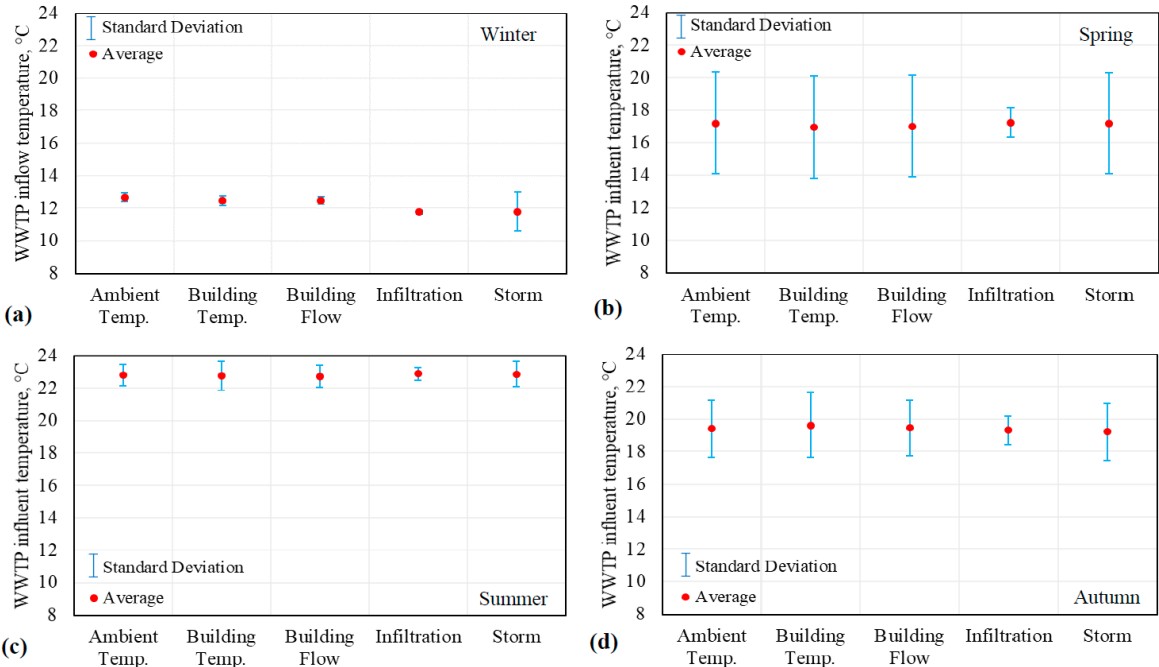

**Figure 6.** Forecasted mean *WWTP influent temperature* value and standard deviations related to the uncertainties of inputs during (**a**) Winter, (**b**) Spring, (**c**) Summer, and (**d**) Autumn.

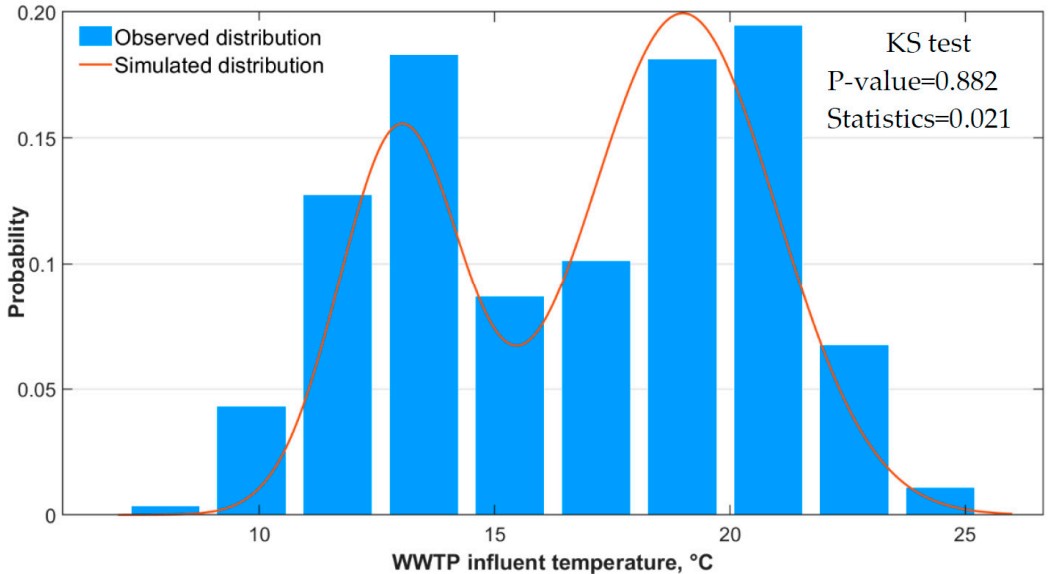

**Figure 7.** Comparison of simulated PDF as Gaussian mixture for *WWTP influent temperature* and distribution of observed data.

The ANN model is used to investigate the impact of 2 °C decrease in building effluent temperature. In that case, *WWTP influent temperature* would decrease from 16 °C to 14.6 °C. Accordingly, 203 GWh year$^{-1}$ heat loss in the sewage network would be avoided due to lower wastewater temperatures. However, since heat recovery in the buildings would reduce the temperature of wastewater at WWTP, heat demand for digesting the sludge in WWTP would be increased by 0.71 GWh year$^{-1}$. Moreover, treated water discharged from WWTP contains less thermal energy, which results in the district heating company recovering 176 GWh year$^{-1}$ less heat from treated water. Table 5 summarises the energy performance of the wastewater system in the current situation and the scenario that 40% of buildings installing heat exchangers. The results show that the ANN model is able to scrutinise the impacts of upscaling local heat recoveries on the performance of prevailing centralised systems.

**Table 5.** Comparison of the current situation and the possible evolving system resulting from local heat recovery by heat exchangers in 40% of buildings.

| Parameter | Current Situation without Local Heat Recovery | Possible Evolving System Regarding Local Heat Recovery |
|---|---|---|
| Buildings effluent temperature, °C | 25.00 | 23.00 |
| WWTP influent temperature, °C | 16.00 | 14.60 |
| Local heat recovery in buildings, GWh year$^{-1}$ | 0.00 | 96.26 |
| Heat loss in sewage network, GWh year$^{-1}$ | 1237.28 | 1034.15 |
| WWTP heat demand, GWh year$^{-1}$ | 11.06 | 11.77 |
| Heat recovery from treated water, GWh year$^{-1}$ | 1485.75 | 1309.95 |

The results of this study indicate that the ANN model can forecast the *WWTP influent temperature* based on historical data commonly recorded with an ignorable error. As mentioned in the literature review, several mathematical models have been suggested for sewage temperature calculation. However, a conflict between the accuracy and simplicity of wastewater temperature models exists in previous studies. For instance, Sonakiya et al. [50] and Abdel-Aal et al. [51] presented a simple model based on temperature loss. However, they reported 0.1 to 4 °C temperature loss per km. This shows that the accuracy of their model varies regarding the place of measurement, and it increases the uncertainty of their model. Abdel-Aal et al. [51] reported the values of root mean square error (RMSE) of their model as 0.9, 0.5 and 0.2 K for three different sites, while the overall RMSE was found to be 0.37 K. The value of RMSE for ANN model in this work is 0.6 K. A more accurate model was proposed by Dürrenmatt and Wanner [16,52]. Their model is a function of 24 parameters related to the sewer pipe, soil, wastewater, and air, whereby eight parameters are more sensitive and should be calibrated by field measurement data. The value of RMSE for the calibrated model was reported as 0.2 K. The ANN model offers an alternative route with fewer required parameters, however still with high accuracy. The ANN model in this work is developed based on historical data commonly recorded in WWTPs without any field measurement. In addition to the simplicity, $R^2$, *RRMSE,* and *PBIAS* values of 0.9456, 0.0548, and 0.0800 show the accuracy of the model.

## 4. Conclusions

The ANN model presented in this work was developed to forecast the *WWTP influent temperature*. The proposed model was trained, validated, and tested using seven input parameters like ambient temperature, building effluent temperature, building effluent flowrate, infiltration into the sewage network, stormwater flow rate, the hour of day, and the day of year. The dataset was related to a period of 10 years, from 2009 to 2018. To investigate the extrapolation ability of the developed network, a dataset related to the year 2019 was prepared as seven inputs to the network, and calculated output was compared to observed values in 2019. The correlation coefficient ($R^2$), relative root mean square Error (*RRMSE*), and percentage bias (*PBIAS*) are used as three statistical indicators to evaluate the performance of the model. The values of 0.945 for $R^2$, 0.055 for *RRMSE*, and 0.080 for *PBIAS* show an appropriate performance of the ANN model.

The Monte Carlo simulation (MCS) technique was also implemented to analyse the relative significance of uncertain input parameters on the output. Sensitivity analysis showed that stormwater flowrate and building effluent temperature are the most influential input parameters all year round on the inflow temperature to the treatment plant. However, sensitivity analysis in different seasons indicated that the significance of parameters is changed. For instance, stormwater is the most influential parameter during winter, while building effluent temperature and ambient temperature are more impactful during summer. Moreover, the MCS sensitivity analysis implemented the ANN model results. The distribution of ANN model results is perfectly similar to the distribution of observed data, which shows that the ANN model is able to forecast the *WWTP influent temperature* efficiently. Our results suggest that ANN as a data-driven model for forecasting is a viable alternative to conventional models, which depend on the physical properties of the system that may be difficult or costly to obtain. Finally, the proposed ANN model used to investigate the effects of upscaling local heat recoveries on the performance of the wastewater system in Stockholm. In the case of installing heat exchangers in 40% of buildings, *WWTP influent temperature* would decrease from 16 °C to 14.6 °C. Accordingly, heat loss in the sewage network would be reduced by 203 GWh year$^{-1}$. However, heat demand for digesting the sludge in WWTP would be increased by 0.71 GWh year$^{-1}$. Moreover, available heat for recovery from treated water by the district heating company would be decreased by 176 GWh year$^{-1}$. In future studies, this model is going to be implemented as an analysis tool to investigate the techno-economic and environmental impacts of local heat and water recoveries on critical centralised functions like WWTP and district heating company in Stockholm.

**Supplementary Materials:** Supplementary materials are available online at http://www.mdpi.com/2071-1050/12/16/6386/s1.

**Author Contributions:** Conceptualization, F.G. and D.N.; methodology, F.G.; software, F.G.; validation, F.G., and V.M.; investigation, F.G. and V.M.; data curation, F.G.; writing—original draft preparation, F.G.; writing—review and editing, D.N. and V.M.; visualization, F.G.; supervision, V.M.; project administration, D.N. All authors have read and agreed to the published version of the manuscript.

**Funding:** This research has been carried out within the SEQWENS project funded by Formas—The Swedish Research Council for Environment, Agricultural Sciences and Spatial Planning, grant number 2018-00239.

**Acknowledgments:** The authors would like to thank Sofia Andersson and her colleagues from Stockholm Vatten och Avfall AB, Sweden, for providing the historical wastewater data needed to develop and validate the model. We are also grateful for comments on an early draft provided by Timos Karpouzoglou and Jörgen Wallin at KTH.

**Conflicts of Interest:** The authors declare no conflict of interest. The funders had no role in the design of the study; in the collection, analyses, or interpretation of data; in the writing of the manuscript, or in the decision to publish the results.

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
