# Peer review of "Forecasting Wastewater Temperature Based on Artificial Neural Network (ANN) Technique and Monte Carlo Sensitivity Analysis"

_sustainability, doi:10.3390/su12166386_

Round 1
Reviewer 1 Report
The paper is interesting and deserves publication, but before some points should be addressed:
- Line 108-108: the sentence "Bylinksy...) seems not be related with the rest of the text. Please consider removing it.
- The Discussion section seems to be a Conclusion section. I recommend to include the most important points of the Discussion in a new section of Results and Discussion.
Reviewer 2 Report
This study proposes a model using neural networks that allows forecasting the water temperature of water treatment plants in relation to their ambient temperature and different flow rates, which, added to a sensitivity analysis, allows for more accurate prediction. The work is novel and shows novelty, however, some recommendations must be addressed before continuing with its processing.
pages 31 to 36: more description of this type of background is required, which is shown when starting the introduction section.
from pages 124 to 128: it must be explained more clearly why this neural network techniques were used as well as the Monte 3 Carlo sensitivity analysis, since they are not based on why the use of the techniques.
I recommend that the diction section be incorporated into the results as part of these and not as a separate section, with the aim of facilitating readers' understanding, pages 374 to 395Review the references given that some do not have the year and / or are not in the magazine format, please recommend not using self-citations only in cases where it is the only alternative
Reviewer 3 Report
Many revision and the result not clear also need compare the result with another method.
Many revision and have to compare the result with another method
Comments from reviewers:
- In section result and discussion not clear, compare the proposed method with other researchers’ methods. There are many previous works. You should emphasize the difference with other researchers’ methods. Add more comparison data.
- The problem definition of this work is not clear. In Sect.1, the drawbacks of each conventional technique should be described clearly. You should emphasize the difference with other methods to clarify the position of this work further.
- The details of the proposed system, such as number of neurons, internal function, etc., are not clear
- Future works as an integral part should be included in the Conclusions
- Abstract and Conclusions. In the abstract and the conclusions is too general, especially at the end where the main results are summarized. (please include some numerical values also)
- What software did you use for estimating the neural network and statistical/graphing work?
Round 2
Reviewer 3 Report
Manuscript is adequate